# Separation of Carbon Dioxide from Real Power Plant Flue Gases by Gas Permeation Using a Supported Ionic Liquid Membrane: An Investigation of Membrane Stability

**DOI:** 10.3390/membranes9030035

**Published:** 2019-03-04

**Authors:** Patrik Klingberg, Kai Wilkner, Markus Schlüter, Judith Grünauer, Sergey Shishatskiy

**Affiliations:** 1Helmholtz-Zentrum Geesthacht, Institute of Polymer Research, Max-Planck-Straße 1, 21502 Geesthacht, Germany; markus.schlueter@preussenelektra.de (M.S.); judith.gruenauer@tesa.com (J.G.); sergey.shishatskiy@hzg.de (S.S.); 2Forschungszentrum Jülich GmbH, Institute of Energy and Climate Research, IEK-1: Materials Synthesis and Processing, D-52425 Jülich, Germany; k.wilkner@fz-juelich.de

**Keywords:** supported ionic liquid membrane, gas permeation, coal-fired power plant flue gas, CO_2_ separation, SO_3_ aerosols

## Abstract

The separation of carbon dioxide from coal-fired power plant flue gases using a CO_2_/N_2_-selective supported ionic liquid membrane (SILM) was investigated and the performance and stability of the membrane during operation are reported. The membrane is composed of a polyacrylonitrile (PAN) ultrafiltration membrane as a support and a selective layer of an ionic liquid (IL), 1-ethyl-3-methylimidazolium bis(trifluoromethylsulfonyl)imide (EMIM Tf2N). The feasibility of large-scale SILM production was demonstrated by the formation of a square-meter-scale membrane and preparation of a membrane module. A flat-sheet envelope-type SILM module containing 0.67 m^2^ of the membrane was assembled. Prior to real flue gas operation, the separation behaviour of the membrane was investigated with single gases. The stability of the SILM during the test stand and pilot plant operation using real power plant flue gases is reported. The volume fraction of carbon dioxide in the flue gas was raised from approx. 14 vol. % (feed) to 40 vol. % (permeate). However, issues concerning the membrane stability were found when SO_3_ aerosols in large quantities were present in the flue gas.

## 1. Introduction

CO_2_ is a greenhouse gas with a significant impact on climate change [1] and a large amount of CO_2_ is emitted to the environment by power plant flue gases. It is very probable that, despite the high risk of global warming, the number of coal-fired power plants in countries with large coal reserves, such as China and the United States of America, will increase in the future [2]. In Germany, the percentage of renewable energies increased from 6% to 29% during the 2000–2017 period [3]. Nevertheless, it is likely that fossil fuel-fired power plants will remain necessary in Germany for many years to ensure a continuous, non-fluctuating energy supply. 

Carbon Capture (CC) technologies afford opportunities to make power plants more climate-friendly [4,5,6]. Gas permeation is one possibility for CC and membranes have several advantages compared with competing processes, for example, amine absorption [7]. Membranes are more load flexible, meaning that they can better deal with changing flue gas compositions. Another argument in favour of gas permeation is the short response time of membranes. This feature makes gas permeation very interesting as a method of CC from flue gases generated by German power plants because, nowadays, some plants start up and shut down several times a year, while others often ramp down into partial-load operation. For high degrees of CO_2_ separation, large membrane areas are required but the separation of CO_2_ from only a fraction of flue gas streams could still be useful since the purified CO_2_ may be used as a feedstock in the chemical industry [8].

Research in the area of CO_2_ separation from gaseous mixtures using gas permeation processes has been attracting increasing attention since the invention of membranes with high CO_2_/N_2_ selectivities (α_CO2/N2_ up to 50 at ϑ = 30 °C) and high CO_2_ permeances (L_CO2_ up to 6 m^3^(STP)/(m^2^·h·bar) at ϑ = 30 °C). These membranes are mainly thin-film composite membranes (TFCMs) with selective layers out of poly(ethylene oxide) (PEO)-containing polymers. One example is a TFCM with a selective layer of PolyActive™ 1500—a commercially available polymer that is based on a poly(ethylene oxide)–poly(butylene terephthalate) (PEO–PBT) block copolymer [9]. Another example is a TFCM with the block copolymer Pebax® MH 1657 (poly(ether-b-amide-6)) [10]. Pebax® has a higher CO_2_/N_2_ selectivity but lower CO_2_ permeance than PolyActive™ 1500 at 30 °C [9]. Also, the second-generation Polaris™ membrane from Membrane Technology and Research, Inc. (MTR) [11], should be mentioned, as it has separation properties comparable to those of the PolyActive™ 1500 TFCM.

Other materials with promising CO_2_/N_2_ separation properties are ionic liquids (ILs). Research in the field of ILs, which are commonly defined as salts with melting points below 100 °C [12], was first published in 1914 by Paul Walden [13]. ILs have properties that make them interesting candidates for the design of new membranes. An important feature of ILs is that they are exceptionally chemically tuneable. An extensive variety of ILs with task-specific properties are currently available. The combination of properties like negligible volatility [14], thermal stability [15], low flammability [16] and high ionic conductivity [17] makes ILs exceptionally interesting for various applications. 

In previous studies, our group investigated three commercially available imidazole-based room-temperature ILs with promising CO_2_/N_2_ separation properties [18,19] as potential materials for the selective layer of gas separation membranes. 1-Ethyl-3-methylimidazolium dicyanoamide (EMIM DCA) has an attractive ideal selectivity (α_CO2/N2_ up to 55 at ϑ = 30 °C). 1-Ethyl-3-methylimidazolium bis(trifluoromethylsulfonyl)imide (EMIM Tf2N) has an ideal CO_2_/N_2_ selectivity of 30 at ϑ = 30 °C. We can see from these examples that the selectivity of some ILs is in the range of values considered characteristic of PEO-containing polymers and the fact that the CO_2_ permeabilities of these ILs [19] exceed those of the aforementioned polymers makes ILs even more appealing candidates for membrane development. Nevertheless, there remains a challenge to develop stable, large-scale (re)producible IL-membranes whose selective IL layer thickness is close to that of PEO-containing polymer membranes.

In recent years, various IL-based membranes for CO_2_ separation have been developed and investigated. Tomé et al. discussed different configurations of IL-based CO_2_ separation membranes from a material engineering point of view [20]. There are several different membrane configurations: supported ionic liquid membranes (SILMs), polymer/IL composite membranes, gelled IL membranes and polymerized ionic liquid (PIL) membranes. Zhou et al. successfully developed a TFCM that was prepared from a room-temperature IL/polymerized (room-temperature IL) composite material [21]. The membrane showed an excellent CO_2_ permeance of 6100 ± 400 GPU (1000 GPU = 2.7 m^3^(STP)/(m^2^·h·bar)) and a good CO_2_/N_2_ selectivity of 22 ± 2. 

However, besides good separation properties, the mechanical and chemical stability of a membrane is also crucial for realizing a long membrane lifetime despite operating in the hazardous environment of a power plant flue gas. An overview of IL-based CO_2_ separation membranes, with a focus on the membrane engineering perspective, was given by Dai et al. [22]. In addition, Wang et al. [23] discussed the chemical stability of ILs. However, to our knowledge, a practical demonstration of whether IL-containing membranes remain stable during their operation in real power plant flue gas has not been reported until now.

To our knowledge, in this paper, the stability of an SILM during the separation of CO_2_ from power plant flue gas is reported for the first time. The SILM used in the study is composed of a polyacrylonitrile (PAN) ultrafiltration membrane as a support and EMIM Tf2N as the IL, which provides the membrane with the selectivity property. This paper reports the preparation of various SILMs that differ in the morphology of the PAN support and the composition of the IL coating solution used for preparation; the results of the experiments are then discussed. The large-scale preparation of the SILM which showed the most promising separation performance after a small-scale preparation is reported. Furthermore, flat-sheet SILM envelopes and a flat-sheet envelope-type SILM module with a membrane area of 0.67 m^2^ were fabricated and the quality control results for each step of the preparation are reported. The membrane separation performance of the SILM for each step of preparation (from SILM sheet to SILM module) was therefore tested with single gases. 

However, the main focus of our research was the investigation of the SILM’s stability during power plant flue gas operation. First, the stability of the SILM was investigated using a gas permeation test stand located at the lignite-fired power plant (block K) of RWE Power AG in Niederaussem, Germany. Second, the SILM module was installed in a gas permeation pilot plant that is located at the hard coal-fired Rheinhafen Dampfkraftwerk in Karlsruhe (block 8) of EnBW AG in Karlsruhe, Germany, where stability investigations were conducted during real flue gas operation. 

## 2. Materials and Methods

### 2.1. Membrane Materials

The ionic liquid (Table 1) 1-ethyl-3-methylimidazolium bis(trifluoromethylsulfonyl)imide, solvents and other chemicals for SILM production were purchased from Sigma-Aldrich Chemie GmbH, Munich, Germany. The chemicals were not further modified before use.

Polyacrylonitrile (PAN) ultrafiltration membranes [26] were produced in-house by the phase inversion process. The PAN porous membranes used as a support for the SILMs were all prepared from PAN–organic solvent solutions using a PAN concentration of 13.4 wt. %. Two different solvent mixtures were used (1 and 2). The composition of the PAN solution in organic solvent, which was used for porous membrane preparation, cannot be disclosed due to the licensing limitations. In addition, the gap width between the doctor blade and nonwoven material during PAN membrane preparation was varied for M1 and M2, each of which had three different variants (a–c). Thus, in total, six PAN membranes (M1a–M1c and M2a–M2c) with different porous layer thicknesses and total porosities were prepared (Table 2 and Figure 1). 

### 2.2. Supported Ionic Liquid Membrane Preparation

This study involved the preparation and characterization of twelve SILMs to investigate the influence of the support structure and the coating solution composition on the membrane performance. The membranes were prepared on PAN ultrafiltration membranes, which were used as the porous support and have proved to be suitable for scalable coating techniques [19]. The support membranes M1a–M1c and M2a–M2c (Figure 1) were coated with two different coating solutions (15 wt. % and 20 wt. % IL in methanol), resulting in twelve SILMs: M1a15–M1c15, M1a20–M1c20, M2a15–M2c15 and M2a20–M2c20. The membranes were prepared with a pilot-scale dip and roller coating machine, which is used in-house for the fabrication of various TFCMs [9,27,28]. Membranes with a length of up to 250 m and a coating width of up to 0.61 m can be prepared. Small-scale dip coating of a porous substrate with an IL–methanol solution was previously described by our group [19]. In contrast to the preparation in the aforementioned study, the preparation in the current study was done without a pore-blocking liquid. During the coating process, the coating solution was applied to the porous support by creating a meniscus between the coating roll and the support, as shown in Figure 2. In the region where the support contacts the coating solution meniscus, both the IL and solvent penetrate the porous support due to capillary force. Immediately after leaving the meniscus, the volatile solvent starts to evaporate and is then forced to evaporate further in a pre-dryer and main-dryer, which are both integrated into the coating machine.

The speed of the coating roll n_roll_ was adjusted to a maximum value of 10 min^−1^ in order to prevent meniscus damage due to the low viscosities of the coating solutions. The membrane was automatically dragged through the coating machine with a constant path velocity of v_path_ = 0.31 m/s. A main-dryer temperature of 60 °C was chosen to reliably evaporate the residual solvent.

### 2.3. Water Permeance Measurements

Pure water permeance measurements were taken to draw conclusions about the regularity of the morphology of the PAN support membranes M1a–M1c and M2a–M2c. The method of measuring water permeance was previously described by our group [19]. The pure water permeance of six circular membrane stamps of each PAN membrane was measured. The water permeance F is the volume V [L] of the water permeating through the membrane with a membrane area A_Mem_ [m^2^] during the time period ∆t [h] at a pressure difference ∆p [bar]:(1)$$F=\frac{V}{A_{\mathrm{Mem}}\cdot \Delta t\cdot \Delta p}.$$

### 2.4. Determination of Single Gas Permeances and Selectivities

Single gas measurements were carried out in order to find the combination of the PAN support (M1a–M1c and M2a–M2c) and coating solution (15 wt.% and 20 wt.%) resulting in the best-performing SILM as judged by the CO_2_ permeance and CO_2_/N_2_ selectivity. For single gas measurements, circular membrane stamps were located in a dead-end testing cell. Round stamps with a membrane area of A_Mem_ = 0.00342 m^2^ were collected from the initial and end parts of each produced SILM sheet (M1a15/20–M1c15/20 and M2a15/20–M2c15/20). The SILMs were impinged by a gauge pressure of $p_{F,i}$ = 2 bar. The permeate pressure during measurements was atmospheric. The resulting steady-state permeate volume flow ${\dot{V}}_{P,i}$ was measured with a Bios Defender 220 flow meter and the permeance L of component i was then calculated according to the following equation:(2)$$L_{i}=\frac{{\dot{V}}_{P,i}}{A_{\mathrm{Mem}}\cdot \left(p_{R,i}-p_{P,i}\right)},$$
where $p_{R,i}$ = $p_{F,i}$ and ideal gas behaviour is assumed due to the low pressures applied. The ideal gas selectivity α_i,j_ is the ratio of the single gas permeances of components i and j:(3)$$\alpha_{i,j}=\frac{L_{i}}{L_{j}}.$$

Furthermore, single gas pressure increase measurements were done with an in-house developed automatic membrane gas transport property determination facility. The facility allows for the automatic determination of the membrane’s gas separation performance for up to 15 gases by utilizing the constant volume/variable pressure method [29] for a feed pressure range of 100–1200 mbar and a temperature range of 5–120 °C. The permeances of O_2_, N_2_ and CO_2_ were determined for the membrane M1c15. In addition, the CO_2_/N_2_ and O_2_/N_2_ ideal selectivities were calculated.

### 2.5. Membrane Envelope Preparation and Testing

Double-sided envelopes of flat-sheet membranes with a diameter of 0.1 m and a total membrane area of 0.0114 m^2^ were manufactured by the thermal welding of two membrane sheets, two nonwoven sheets and a permeate spacer sheet for each envelope. The sheets were prepared with a punch. The outsides of the membrane envelopes are formed by SILM sheets, with the selective layer outside (see Figure 3). The permeate spacer is located in the middle of the envelope to ensure a permeate channel with sufficient thickness during operation. In order to prevent membrane damage from the rough spacer, a nonwoven material is placed between the SILM sheet and the spacer on each side.

Following the preparation, membrane envelope quality control testing was done with an in-house developed envelope-testing unit. With the envelope-testing unit, the permeances of the single gases (N_2_, O_2_ and CO_2_) were measured and CO_2_/N_2_ and O_2_/N_2_ selectivities were calculated for each envelope. Finally, membrane envelopes with an unsatisfactory permeance or selectivity were rejected and intact ones were integrated into module housing. 

### 2.6. Module Preparation and Testing

An envelope-type [30,31,32] K100 flat-sheet module (membrane module with an inner diameter of 100 mm and maximum membrane area of 1 m^2^ (see Figure 4) was prepared by installing 59 SILM envelopes for a total membrane area of 0.67 m^2^. After module preparation, the quality of the module was checked to exclude leaks that formed during module manufacturing. For quality control, the module was pressurized at a defined feed pressure with compressed air. The resulting permeate volume flow was measured with a Bios Defender 220 flow meter (Mesa Laboratories, Inc., Lakewood, CO, USA) and the oxygen volume fraction in the permeate was measured with a SICK Transic111LP oxygen meter (SICK AG, Waldkirch, Germany). The O_2_ and N_2_ permeances and O_2_/N_2_ selectivity then were calculated using Equation (2). To make a comparison, the permeance and selectivity determination was done again using the single gases (O_2_ and N_2_).

### 2.7. Investigation of SILM Stability during Real Power Plant Flue Gas Operation

To investigate the stability of the SILM during real power plant flue gas operation, the membranes were tested at two power plants. The first site was a gas permeation test stand located at the lignite-fired power plant Niederaussem (block K) of RWE Power AG, Bergheim, Germany. The second location was the hard coal-fired Rheinhafen-Dampfkraftwerk (block 8) of EnBW AG in Karlsruhe, Germany, where a gas permeation pilot plant operates.

#### 2.7.1. Gas Permeation Test Unit at Lignite-Fired Power Plant

The tests in the flue gas of the lignite-fired power plant were performed using a small-area membrane gas permeation test rig. This test rig was designed to test membrane samples with areas between 8 cm² and 50 cm² at temperatures between 40 °C and 70 °C and water content between 2% rH and 80% rH. The test stand can be used to conduct stability tests of membrane samples before building a large-area module. The test rig is located at RWE Power AG’s “Coal Innovation Centre” at the power plant Niederaussem in Germany and operated by the Forschungszentrum Jülich IEK-1. The test rig is installed downstream of a high-performance scrubber (FGDplus)—an optimized flue gas scrubbing pilot plant [33]. Figure 5 shows a simplified flowchart of the gas permeation test rig. Before entering the membrane test cell, the flue gas is cooled in K1 to a desired dew point temperature and superheated in K2 to the test temperature. By these pre-treatment steps, the relative humidity of the feed gas was defined as 35% relative humidity (dew point = 22 °C) and damage to the membrane material from condensation in the test cell should be avoided (see also 3.6.2). After the filter (F1), a feed pump (P1) was installed to overcome the feed-/retentate-side pressure drop and to adjust a constant feed volume flow. The feed volume flow during the SILM measurement campaign was about 0.12 m³(STP)/h and the feed pressure was about 1.06 bar. The feed temperature was kept nearly constant at ϑ_F_ ~40 °C by trace heating the pipes and the heated test cell. The driving force was generated by a membrane vacuum pump (P2), which reduced the permeate pressure to 20 mbar.

#### 2.7.2. Gas Permeation Pilot Plant at Hard Coal-fired Power Plant

The SILM stability was also investigated during operation at the hard coal-fired Rheinhafen-Dampfkraftwerk by EnBW AG in Karlsruhe, Germany. The SILM module was therefore installed in a gas permeation pilot plant (see Figure 6). The pilot plant, which can be run in one-stage parallel mode and in two-stage mode, was previously described by Pohlmann et al. [34]. In contrast to the aforementioned study, the arrangement of the side channel blower (C1) was reconfigured so that the membrane module M1.1 was on the suction side of the blower. This reconfiguration was to prevent the deposition of particulate matter and acidic condensates within the blower housing.

In order to avoid flue gas condensation within the pilot plant—more specifically, within the module housing—the pilot plant was purged with ambient air for at least one hour before and after flue gas operation. During flue gas operation, a small fraction of the power plant off-gas was sucked out of the power plant chimney into the pilot plant. To avoid membrane damage by acidic condensates during operation, the flue gas was permanently cooled in the condenser H1 to ϑ_1_ ~22 °C and constantly superheated in the electrical heater H2. The superheating resulted in a nearly constant feed temperature of ϑ_F_ ~33 °C throughout the entire measurement campaign. The feed standard volume flow ${\dot{V}}_{N,F}$, which is regulated by the rotational speed of C1, was also kept constant (${\dot{V}}_{N,F}$ ~5.5 m^3^(STP)/h). For a sufficient driving force, the permeate pressure $p_{p}$, which is regulated by a pressure control valve, was continuously adjusted to 125 mbar. 

### 2.8. Process Simulation

Steady-state process simulations were run for an indication of the SILM stability during operation at the hard coal-fired power plant. For this purpose, the equation-oriented simulation software Aspen Custom Modeler^®^ [35] was used. In-house developed models for the phase separator, heat exchanger, pumps and membrane modules were used to simulate the gas permeation process. 

The simulation model of the membrane module was described in detail by Brinkmann et al. [31]. Geometrical parameters were adjusted according to the dimension of the K100 module. Real gas behaviour, concentration polarization and non-isothermal behaviour were considered, as well as the pressure drop in the feed/retentate and permeate channel. An adjustment to the permeate-side pressure drop equation was necessary to achieve a better match between the simulation and experimental results. For the calculation of gas properties, the cubic Soave–Redlich–Kwong equation of state extended for the better treatment of water vapor according to Kabadi and Danner (SRKKD) [36] as implemented in Aspen Properties used. The permeance L of component i at temperature T was calculated using the model of a membrane module according to the Arrhenius equation:(4)$$L_{i}=L_{\mathrm{fac}}\cdot L_{\infty ,i}^{0}\cdot \exp\left(-\frac{E_{i}}{R\cdot T}\right),$$
where R is the universal gas constant. The activation energy E_i_ and the permeance of component i at infinitely high temperature $L_{\infty ,i}^{0}$ were determined using the results of the single gas membrane performance test of the corresponding membrane. The scaling factor L_fac_ was calculated in order to adjust the permeance calculations. This is necessary if differences in the gas permeances of the initial measured SILM stamp (measured with the pressure increase method) and the final SILM module (module quality control results) are found.

For every steady-state run of the process simulation, the measured values of feed temperature ϑ_F_, feed standard volume flow ${\dot{V}}_{N,F}$, feed pressure $p_{F}$, feed mole fraction $x_{F,i}$ (i = CO_2_, N_2_, O_2_ and H_2_O) and permeate pressure $p_{P}$ were set as fixed variables. Trace substances like SO_x_ and NO_x_, as well as solid and liquid particles (aerosols), were neglected.

## 3. Results and Discussion

### 3.1. Investigation of PAN Supports and Coating Solution Influence on SILM Quality 

The main aim of the conducted study was to investigate the stability of an SILM during real flue gas operation. To get a mechanically stable SILM that also has in addition a good membrane separation performance, twelve membranes (M1a15/20–M1c15/20 and M2a15/20–M2c15/20) were prepared. The most promising membrane in terms of separation performance was chosen for large-scale SILM production and stability investigation with real power plant flue gas. In this section, the results of the PAN support investigation and SILM separation performance investigation are presented and discussed.

#### 3.1.1. PAN Support Investigation

The stability of an SILM is highly dependent on the morphology of the support membrane. It is assumed that the presence of macro-voids in the support leads to unwanted imperfections in the IL layer and can also result in mechanical damage to the membrane during its use. As can be seen in Figure 1, the six PAN membranes did not have any macro-voids. 

Water permeance measurements were taken to determine the regularity of the support membrane morphology. The M1c membrane had the smallest standard deviation for water permeance (see Figure 7); this could be an indication that the M1c support had the most regular morphology over the membrane length. A support membrane with a regular morphology throughout the whole batch is a promising candidate for the preparation of a highly selective, mechanically stable SILM membrane, because imperfections in the IL layer are not likely to occur during selective layer formation.

Comparing the membranes made with the same solvent composition (M1 or M2), it can be seen that the water permeance decreased with increasing PAN membrane thickness. Since all membranes were made with the same amount of PAN (13.4 wt. %) and the M2 membranes were thinner than M1, the M1 supports had a higher overall porosity compared with M2. This was also confirmed by the fact that the water permeances of the M1c and M2a membranes were similar, although M1c had a greater thickness than M2a. The overall porosity of the support substructure can influence the membrane performance of the resulting SILM in several ways. On the one hand, a dense matrix-structure can prevent the IL from penetrating the support too deeply during coating. This would result in a thinner IL layer and, therefore, a higher permeance of the SILM. On the other hand, comparing two SILMs with the same IL thickness, a dense substructure and/or high tortuosity of the support will decrease the membrane permeance, since the permeation distance for the gases (effective selective layer thickness) is higher.

Supported liquid membranes prepared using porous membranes with narrow surface pore size distributions are less sensitive to pressure fluctuations [37]. Furthermore, when using a support with a narrow surface pore size distribution, it is likely that the prepared SILM will have fewer imperfections in the IL layer. From the SEM images (Figure 1), it can be seen that the surfaces of the PAN membranes M1a–M1c and M2a–M2c are similar in morphology. The pores were counted and there were approximately one thousand pores per square micrometre, with a pore diameter of 6 ± 1.86 nm. In terms of surface porosity, isoporous membranes made from PS-*b*-P4VP seem to be better suited for SILM production [18]. Nevertheless, PAN membranes were chosen for SILM production, since PAN membranes can be produced at a large scale, are chemically and mechanically stable for the application at hand, and, therefore, are suitable for scalable membrane preparation.

#### 3.1.2. SILM Investigation

The support membranes M1a–M1c and M2a–M2c, discussed in the previous section, were each coated with two different coating solutions (15 wt. % and 20 wt. % IL in methanol). The twelve resulting SILMs were then subjected to single gas measurements. In this section, the dependence of the resulting selectivities and permeances on the support properties and coating solution properties is briefly discussed.

First, the single gas measurement results showed that using a coating solution with a lower IL concentration resulted in more permeable SILMs (Figure 8). It is probable that a higher concentration of IL in the coating solution leads to a greater pore volume over the membrane thickness being filled with the IL, what in turn results in lower permeances.

Secondly, comparing the membranes M1 and M2, it can be seen that the coating solution composition had different effects on the selectivity. The M1 membranes were more selective when coated with the 15 wt.% coating solution. The opposite was found for the M2 membranes, for which the 20 wt.% coating solution led to more selective SILMs (Figure 9).

A conclusive explanation or prediction for the above observation and for the dependence of the selectivities and permeances on the overall porosity of the support membranes was not found. This is likely due to the fact that the properties of the porous support, as well as the properties of the coating solutions, can influence the SILM performance in positive and/or negative ways. This was briefly discussed regarding the influence of the overall support membrane porosity on the resulting permeance of the SILMs (see Section 3.1.1). There is another proposed mechanism that could explain the varying effects of IL concentration in the coating solution. On the one hand, due to the lower viscosity, less concentrated coating solutions are likely to more deeply penetrate the pores of a support. On the other hand, this does not automatically result in a lower permeance: since the amount of IL is lower, it therefore fills less pore volume.

Our primary aim for this study was to investigate the SILM stability during real flue gas operation. We decided to prepare M1c15 on a large scale, because this variant showed the best overall membrane performance.

### 3.2. SILM Large-Scale Preparation and Quality Control: From SILM Sheet to SILM Module

Prior to the stability investigations of SILMs operating in power plant flue gas, the M1c15 membrane was prepared on a pilot scale. An SILM sheet having a membrane area of about six square meters was prepared. From that batch, 59 intact membrane envelopes were prepared and incorporated into a membrane module having a total membrane area of 0.67 m^2^. In this section, the results of the pressure increase measurements, as well as the results of quality control testing after each step of preparation (from membrane sheet to module), are presented and discussed.

#### 3.2.1. “Pressure Increase” Measurements

Figure 10 shows the temperature depending permeances of CO_2_, O_2_ and N_2_ of the pilot scale produced M1c15 SILM in the form of an Arrhenius plot. The data were determined with the “pressure increase” (constant volume/variable pressure) method. A temperature range of 21–67 °C was chosen for the investigation, because this is a typical range for the feed temperature during flue gas operation (ϑ_F_ ~33–40 °C). As can be seen, within the investigated temperature range, the permeance of the measured components increased with increasing temperature. However, no conclusions about the dependence of permeance on temperatures outside of the investigated range should be drawn from these results, since this trend can change at other temperatures. This is due to the fact that the permeation of a gas is mostly influenced by sorption to and diffusion through the ionic liquid and the two processes exhibit opposing temperature dependencies. Furthermore, the permeability can also be influenced by material flow through open pores, since imperfections in the IL layer are likely to be present in the SILM M1c15.

The mean pressure between feed and permeate during the pressure increase measurements was chosen to be about 250 mbar for all gases. One has to keep in mind that, during real flue gas operation, the mean pressure is different for each component since the mole fraction of each component is different in the feed stream, as well as in the permeate stream. In addition, the total mean pressure slightly fluctuates during operation, although $p_{F}$ and $p_{P}$ were adjusted to a constant value. The dependence of the permeances on the mean pressure was not investigated in detail and hence neglected in process simulation. The temperature-dependent permeance was calculated in the simulation model according to the Arrhenius equation (Equation (4)). The Arrhenius model parameters (Table 3) were determined from the pressure increase results. 

The results of the pressure increase measurements showed that the previous selectivity and permeance values (i.e., the measurements attained during the PAN support and IL-coating investigation campaign for SILM M1c15 (α_CO2/N2_ = 28.35 ± 0.51 and L_CO2_ = 2.00 ± 0.35 m^3^(STP)/(m^2^·h·bar); Figure 8 and Figure 9)) could not be achieved again after the large-scale preparation of M1c15. This reveals issues concerning the reproducibility of a qualitatively equal SILM. Nevertheless, the results show that it is possible to prepare an SILM on a large scale with a pilot-scale coating machine and a porous PAN support. 

#### 3.2.2. SILM Envelope and Module Preparation

SILM envelopes were prepared from the M1c15 large-scale batch and quality control testing of these envelopes was carried out using single gas measurements. Then, satisfactorily performing envelopes were selected and packed into a K100 module housing. Lastly, quality control testing of the SILM module was performed with single gases and compressed air. 

In total, 79 membrane envelopes were welded. Of this group, 59 intact envelopes were chosen for module preparation. In Table 4, the average results of single gas SILM envelope quality control measurements are reported. 

The main factor driving envelope selection was CO_2_/N_2_ selectivity because it enables a clear observation of CO_2_ enrichment of the permeate during flue gas operation; this, in turn, gives a strong indication of membrane stability. An average CO_2_ permeance of 1.53 m^3^(STP)/(m^2^·h·bar) at 20 °C was assumed to be sufficiently high. The average CO_2_ permeance of the envelopes revealed that the large-scale SILM batch had irregularities in IL thickness, since a standard deviation of 0.35 m^3^(STP)/(m^2^·h·bar) was determined and a permeance of 1.72 m^3^(STP)/(m^2^·h·bar) at 20 °C was calculated using the Arrhenius parameters (Table 3).

After preparation of the SILM module, the permeances of O_2_ and N_2_ were determined with both compressed air and single gases at ϑ = 25 °C and ∆p = 2 bar (Table 5) for quality control. Testing with CO_2_ was not possible since the CO_2_ permeate volume flow was out of the measuring range of our Bios Defender 220 flow meter. 

Comparing the results of SILM module quality control and SILM envelope quality control, it can be seen that the module preparation did not introduce any leakages to the module. Differences in O_2_/N_2_ selectivities, when using compressed air or single gases, can be simply explained by the sensitivity of the selectivity calculation, since small inaccuracies during measurements cause large deviations in selectivity calculations. 

For the process simulation, the factor L_fac_ = 0.81 was calculated by dividing the N_2_ permeance calculated with the Arrhenius equation at 25 °C and the N_2_ permeance determined with the compressed air module quality control results. The scaling factor (see Equation (4)) was then included in the simulation model as a fixed variable. 

### 3.3. SILM Stability Investigation during Flue Gas Operation

In this section, the results of the SILM stability investigation during real power plant flue gas operation are presented. The separation properties of the membrane before and after flue gas operation are reported and discussed. Differences in flue gas compositions at both experimental sites are described and related to the SILM stability investigation results. 

#### 3.3.1. Gas Permeation Test Stand Investigation at Lignite-Fired Power Plant

Two experiments with samples of the SILM M1c15 material were carried out at the power plant in Niederaussem. Figure 11a shows a membrane sample before exposure; Figure 11b shows sample 1 after 113 h of operation; and Figure 11c shows sample 2 after 335 h of flue gas operation. The membrane area of the samples is 8 cm². When removing sample 1 (Figure 11b), parts of the membrane became stuck to the test cell sealing ring; apart from this, there were no visible changes in the samples caused by flue gas exposure.

The data from the experiment with sample 1 are shown in Figure 12. The test lasted for 113 h and the experiment was performed under flue gas conditions for the first 70 h (before the FGDplus system ceased operation).

The measurement data from the test stand in the power plant did not indicate deterioration in the membrane performance during flue gas operation (see Figure 12 and Figure 13). The fluctuations in CO_2_ concentration in the permeate (red line) are the result of the low permeate flow causing unstable measured values. This observation is, however, not critical to the qualitative assessment of the SILM stability. The constant increase in CO_2_ concentration from a volume fraction of 0.09 (Feed) up to a volume fraction of 0.35 (Permeate) for the first 70 h proves the stability of the membrane under flue gas conditions.

The second test was carried out with a new membrane stamp (sample 2) for 335 h with a continuous supply of flue gas (Figure 13). Again, there was considerable fluctuation in the CO_2_ permeate concentration caused by the permeate flow being too low. By comparing tests 1 and 2, the lower concentration of CO_2_ in the permeate (an average volume fraction of 0.28) can be explained by the lower CO_2_/N_2_ selectivity of sample 2 (see Table 6). An explanation for the slight increase in the CO_2_ concentration in the permeate during the measurement campaign is rather speculative and could be due to the membrane or measurement inaccuracy. Overall, the results show no degradation in membrane performance during the flue gas operation under the test conditions.

Aiming to derive a quantitative expression of the membrane performance before and after the flue gas exposure, single gas permeation tests were performed on the two samples. The results are summarized in Table 6 and show significant differences in single gas selectivity and permeance before and after contact with the flue gas. These disparities contradict the results obtained from the test stand experiments and can only be explained by the changes occurring after the end of the flue gas exposure. The changes in the membrane permeances and selectivities observed after flue gas exposure may be due to a loss of IL from the matrix-structure since the selectivity decreased and the CO_2_ and N_2_ permeances increased. 

#### 3.3.2. Gas Permeation Pilot Plant Investigation at Hard Coal-Fired Power Plant

To investigate the SILM stability, the SILM module was installed in the gas permeation pilot plant (Figure 6), which is located at the EnBW hard coal-fired power plant in Karlsruhe, Germany. 

Figure 14 shows the CO_2_ volume fraction in the feed and permeate, as well as the feed temperature, measured during the first flue gas operation period. The percentage of CO_2_ increased from approx. 14 vol. % in the feed to an average value of 35 vol. % in the permeate. It can be observed that the SILM performed stably during the first flue gas operation period. This can be concluded because, considering the trend for $y_{F,{\mathrm{CO}}_{2}}$, and ϑ_F_, the trend for y_P,CO2_ is reasonable. During the first 9 h of flue gas operation, the volume fraction of CO_2_ in the permeate decreased due to the decreasing CO_2_ volume fraction in the feed and the increasing feed temperature. After approx. 14 h, the volume fraction in the feed increased, which, in turn, resulted in an increasing CO_2_ volume fraction in the permeate. A further indication of the stable performance of the SILM module during the first operation period is the good match between the experimental and simulation results of $y_{P,{\mathrm{CO}}_{2}}$. 

After approx. 17 h of flue gas operation, the investigation had to be stopped due to power plant downtime, which lasted for 17 days. After 432 h, the SILM module stability investigation resumed. 

From the beginning of the second flue gas operation period, the volume fraction of CO_2_ in the permeate constantly decreased until t = 462 h, at which time a nearly constant CO_2_ volume fraction in the permeate was observed. The gap between experimental and simulation results increased, illustrating that the SILM module was damaged during the second flue gas operation period at the hard coal-fired power plant in Karlsruhe. Issues regarding feed concentration measurements from t = 432 h to t = 442 h have to be mentioned, since such high values for $y_{F,{\mathrm{CO}}_{2}}$ (up to 0.2 vol/vol) are not realistic in the flue gas of coal-fired power plants (see Figure 15). This also explains why the experimental and simulation results do match well at the beginning of the second flue gas operation period.

After the measurement campaign, the SILM module was removed from the pilot plant and follow-up examinations were performed to investigate possible causes of membrane damage. The SILM envelopes were removed from the module housing and the average relative change in single gas permeances after flue gas operation was determined (Table 7). 

Unlike the SILM samples measured at the lignite-fired power plant, these samples did not show a loss of IL after flue gas exposure. This is because the average CO_2_ permeance was lower, whereas the average permeances of O_2_ and N_2_ were higher after flue gas operation. This agrees with an average CO_2_/N_2_ selectivity of 6.6 ± 3.5 and an average O_2_/N_2_ selectivity of 1.1 ± 0.1. 

In order to avoid membrane damage, on the one hand, the pilot plant was always purged for one hour with ambient air before and after flue gas operation. On the other hand, a drop in flue gas temperature to below its dew point temperature in the membrane module was prevented by the pilot plant’s pre-treatment steps (see Figure 6). The flue gas was cooled down in H1 and subsequently superheated in H2 before entering the membrane modules M1.1 and M1.2 (in one-stage parallel mode). Hence, the membrane damage due to the deposition of acidic condensates on the membrane surface can be excluded. To draw conclusions about the cause of membrane damage during flue gas operation in Karlsruhe, the differences in the power plants’ off-gas composition at both experimental sites were compared.

### 3.4. Comparison of Flue Gas Compositions at Both Experimental Sites

The SILM performed stably during flue gas operation for at least 335 h at the power plant in Niederaussem, whereas the membrane was quickly damaged during operation at the power plant in Karlsruhe. The composition of a power plant’s off-gas is influenced by the fuel, combustion conditions and flue gas cleaning steps. In this section, the differences in coal and flue gas cleaning at both power plants are reported and discussed. The results uncover the reason for the nearly immediate damage of the SILMs when operating at the power plant in Karlsruhe.

First of all, it has to be mentioned that the RWE power plant in Niederaussem is lignite-fired, whereas the EnBW power plant in Karlsruhe is hard coal-fired. Usually, lignite has a higher sulphur content compared with hard coal; consequentially, the concentration of SO_x_ in the flue gas of lignite-fired power plants is higher than that in hard coal-fired power plants when comparing non-cleaned flue gas streams. The Rhenish lignite has a relatively low concentration of sulphur compared with lignite from other sources, so conclusions about the differences in SO_x_ concentrations in both flue gases due to the coal types would be vague. In addition, the quality of the coal at both power plants slightly changes from time to time, so a conclusion about membrane damage due to coal quality is hardly possible. 

The composition of the power plant off-gases is influenced by the different flue gas cleaning steps. Both power plants are equipped with standard cleaning steps for particulate removal (dry electrostatic precipitator (ESP)) and removal of sulphur oxides (wet flue gas desulfurization (WFGD)). The WFGD is used in general at both power plants; however, at RWE Niederaussem, a high-performance FGD pilot plant [33] (FGDplus pilot plant from ANDRITZ Energy & Environment) is under investigation and was in operation during the SILM measurement campaign upstream of the gas permeation test stand. The FGDplus pilot plant is constructed modularly and consists of a total of five modules (counter flow module, FGDplus module, spray tower module, droplet separator module and wet electrostatic precipitator module (WESP)) [38]. The concentration of SO_x_ in the power plant off-gas is further reduced by the FGDplus pilot plant compared with the generally used WFGD. Nevertheless, this does not seem to be the key difference between the two power plants, which is decisive for stable membrane performance. The main difference in general flue gas cleaning between the RWE power plant (block K) in Niederaussem and the EnBW power plant (block 8) in Karlsruhe rather lies in the way that nitrogen oxides are avoided/reduced. At the RWE power plant in Niederaussem, the concentration of NOx is held under its legal limit by using combustion measures, whereas a selective catalytic reduction (SCR) with ammonia as the reducing agent is used in Karlsruhe.

Our research led us to a problem that was observed for post-combustion amine absorption processes [38,39,40]—the presence of solid and liquid aerosols in power plant flue gases. Liquid SO3 aerosols are known to form in the flue gas of sulphur-containing fuels. The formation of small sulfuric acid droplets with a diameter ≤ 10 µm, also termed mist or fog, was previously described by Schaber (1995) [41]. Water vapor and SO_3_ in the flue gas react to form gaseous H_2_SO_4_. In addition, we know from Li et al. [42] that the ammonia (NH_3_) used in an SCR can react with H_2_O and SO_3_ to generate ammonium sulphate ((NH_4_)_2_SO_4_) and/or bisulphate (NH_4_HSO_4_). The rapid drop in temperature in the WFGD leads to the supersaturation of all SO_3_ molecules and homogeneous liquid and/or heterogeneous solid–liquid aerosols are formed by nucleation mechanisms [43]. Most of the SO_3_ molecules (H_2_SO_4_, (NH_4_)_2_SO_4_, NH_4_HSO_4_) stay in the gaseous state until they reach the WFGD. In cooler regions of the flue gas flow path, condensation of the SO_3_ molecules can even occur before entering the WFGD. Comparing the flue gas cleaning steps of both power plants and their influence on aerosol formation results in a reasonable explanation for the membrane damage in Karlsruhe compared with the stable SILM operation in Niederaussem (see Table 8)—the presence of larger quantities of SO_3_ aerosols, such as sulfuric acid and especially sticky ammonium sulphate and/or bisulphate, in the EnBW power plant (block 8) in Karlsruhe. In particular, the formation of sticky ammonium sulphate and bisulphate aerosols due to the presence of ammonia as a reducing agent in an SCR [42] is possible at the EnBW power plant in Karlsruhe. In addition, it is likely that more gaseous H_2_SO_4_ formed in Karlsruhe since the additional oxidation of SO_2_ to SO_3_ may occur in an SCR [42]. Additional SO_3_ may have also formed in the ESP [38], which applies to both power plants. Another factor for aerosol formation, which was not proven by us, may be the temperature difference during the desulfurization step, since the homogeneous (liquid) and heterogeneous (solid–liquid) SO_3_ aerosols form in the WFGD. A smoother cooling of the flue gas within the WFGD probably results in fewer SO_3_ aerosols being formed. The flue gas at the RWE power plant cools from 105 °C to 67 °C [38], whereas the flue gas at the EnBW power plant undergoes sharper cooling, from 140 °C to 65 °C (see Table 8). 

Another argument for small-size droplets (nanometre scale) being the cause of membrane damage is that aerosols are hardly separated by the pilot plant’s pre-treatment steps. During the previous PolyActive™ measurement campaigns [34], where the blower C1 was arranged upstream of the membrane modules M1.1 and M1.2 (in one-stage parallel mode (see Figure 6), membrane damage as described in this publication was not observed. The membrane showed a stable performance but issues with gypsum particles and acidic depositions within the blower housing were observed. It was decided to install the blower C1 downstream of the membrane modules for the current measurement campaign. That, in turn, probably prevented the aerosols from becoming stuck in the blower housing, so they were deposited in the membrane module. Because of the sharp changes in the flow directions and feed spacers within the module housing [31], the especially sticky (NH_4_)_2_SO_4_ and/or NH_4_HSO_4_ aerosols could have deposited on the membrane surface. Another possible reason for the stable performance of the PolyActive™ module in the previous measurement campaign [34] is that the pilot plant was operating at block 7 at the EnBW power plant during the previous measurement campaign and it is very likely that smaller quantities of aerosols were in the off-gas of block 7. This is because, on the one hand, low-sulphur coal was burned during our measurement campaigns at block 7. On the other hand, the flue gas was reheated before entering the corrodible chimney of block 7.

To underpin the presumption that the SILM damage was caused by SO_3_ aerosols, the SILMs that were used in Karlsruhe were analysed for the presence of NH_4_^+^ and SO_4_^2−^. Therefore, eight stamps (A_total_ = 49.3 cm^2^) of the flue-gas-exposed SILMs and non-flue-gas-exposed SILMs were each washed for one week with 100 mL of ultrapure water at 50 °C. The concentrations of NH_4_^+^ were then photometrically determined by the EnBW laboratory staff and the concentrations of SO_4_^2−^ were determined by ion chromatography. The results are shown in Table 9.

Low concentrations of SO_4_^2−^ (c(SO_4_^2−^) < 0.1 mg/L) in the washing solutions were found. However, the performed tests revealed clearly that a higher amount of ammonium was deposited on the membranes after flue gas exposure. For additional evidence on the accumulation of acidic species on the membrane surface, three of the flue-gas-exposed SILM envelopes (Figure 16) were dried under vacuum for 15 h at 50 °C and subsequently wetted with distilled water. The pH value was then determined using litmus paper (Figure 16 (top)). The acidic liquid (pH value = 1–2) was verified to contain SO_4_^2−^ on the basis of its white turbid appearance (formation of barium sulphate (BaSO_4_)) upon being mixed with barium chloride solution (BaCl_2_). No visible change was observed when being mixed with sodium chloride, ammonium chloride or calcium chloride solution (Figure 16 (bottom)). 

The litmus test was repeated with the SILMs that were used in Niederaussem, block K. The applied liquid at the surface had a pH value of 7, just like the results of the same test with non-flue-gas-exposed SILMs. Therefore, the possibility of acidic species accumulating on the surface of the SILMs operating in Niederaussem can be excluded. The final investigation established that SO_3_ aerosols are very likely the cause of the SILM damage during flue gas operation at the EnBW power plant (block 8) in Karlsruhe, since SO_4_^2−^ and NH_4_^+^ were proven to be deposited on the membranes.

To summarize, it can be concluded that the presence of aerosols in power plant off-gases not only seems to be an issue for post-combustion absorption processes but is also a general problem for post-combustion carbon capture technologies. Na_2_SO_4_ aerosol with very small solid particles (characteristic particle size d_P_ < 50 nm) appears to be the main cause of amine loss at the amine-based post-combustion absorption pilot plant in Niederaussem but effective countermeasures have been reported [39]. Liquid and sticky SO_3_ aerosols also cause amine losses but are especially a problem for post-combustion gas permeation processes when present in large quantities in the power plant off-gas, since the membrane can be blocked or chemically degrade.

## 4. Conclusions

A method to successfully transfer the production of an SILM from the laboratory to pilot scale is presented in combination with a thorough investigation of the required quality of the employed porous support membrane. As far as we know, this is the largest area of an SILM (A_Mem_ ~6 m^2^) to be prepared and subsequently mounted into a test cell and pilot plant membrane module (A_Mem_ = 0.67 m^2^).

The SILM was tested under real-life flue gas conditions at lignite- and hard coal-fired power plants. The volume fraction of carbon dioxide in the flue gas was raised from approx. 14 vol. % (feed) to 40 vol. % (permeate).

The results show that the separation of CO_2_ from power plant flue gases by gas permeation using SILMs is worthy of consideration and investigation in the future, since the stability of an SILM persisted for at least 335 h when sticky (NH_4_)_2_SO_4_ and/or NH_4_HSO_4_ aerosols were absent and H_2_SO_4_ aerosols in low quantities in the flue gas. The influence of SO_3_ aerosol formation on the power plant cleaning steps is summarized. The presence of SO_3_ aerosols is a general problem for CC technologies. When an SCR for the reduction of NO_x_ is used, special efforts ought to be undertaken, either by removing existing aerosols or by preventing the formation of aerosols. 

Nevertheless, keeping the investment costs for the membranes in mind, an SILM lifetime longer than 250 h is to be proven. In addition, to date, the membrane performance of PEO-containing polymer membranes, like PolyActive™ and Polaris™, exceeds that of the SILM used in this study. However, there is potential for high-permeance membrane development [21] when using ILs for gas-separating membranes. Figure 17 shows a CO_2_/N_2_ selectivity versus CO_2_ trade-off plot where the enclosed area represents the region of permeability versus selectivity which is desirable for CO_2_ separation from flue gas as suggested by Merkel et al. [44]. Inside this region the pressure ratios, required membrane area and feed side gas velocities needed for good membrane performance are in an economically viable regime.

## Figures and Tables

**Figure 1 membranes-09-00035-f001:**
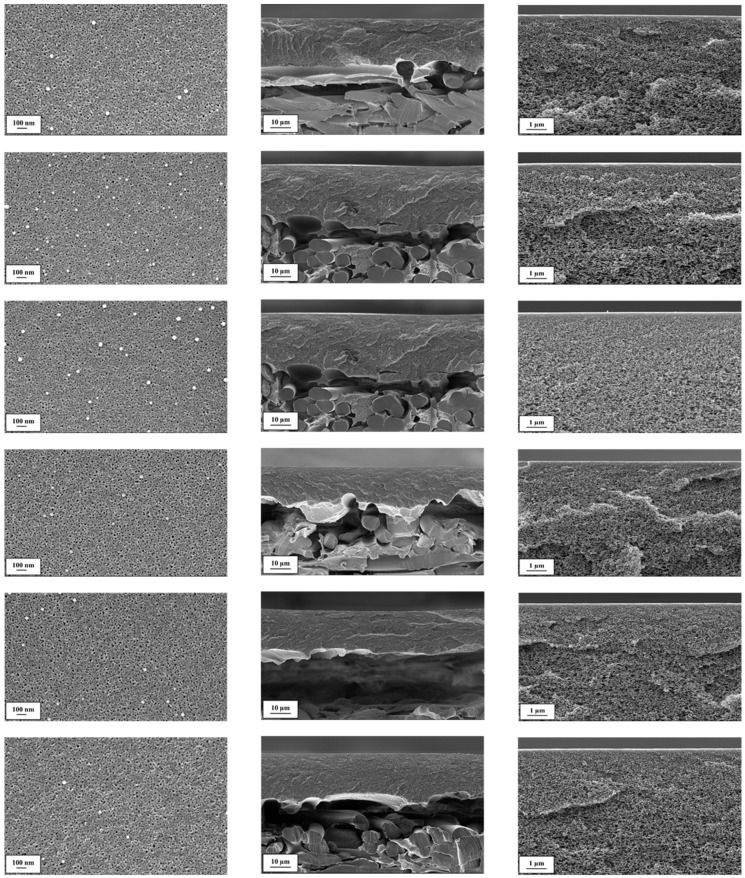
Scanning electron microscope (SEM) images of PAN membranes M1a, M1b, M1c, M2a, M2b and M2c (top to bottom) of (**left**) membrane surface, (**middle**) membrane cross-section and (**right**) membrane cross-section at higher magnification.

**Figure 2 membranes-09-00035-f002:**
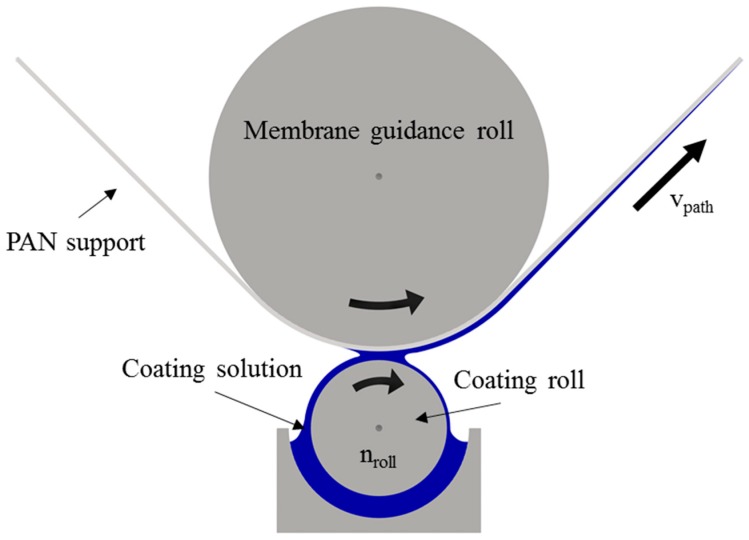
Drawing of the two-roll system of the large-scale dip and roller coating machine. Meniscus formation between the PAN support and coating roll.

**Figure 3 membranes-09-00035-f003:**
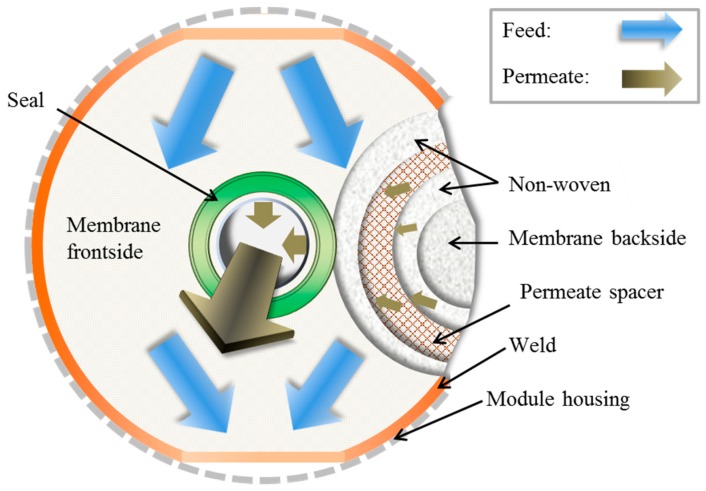
Drawing of membrane envelope with different layers with flow directions indicated.

**Figure 4 membranes-09-00035-f004:**
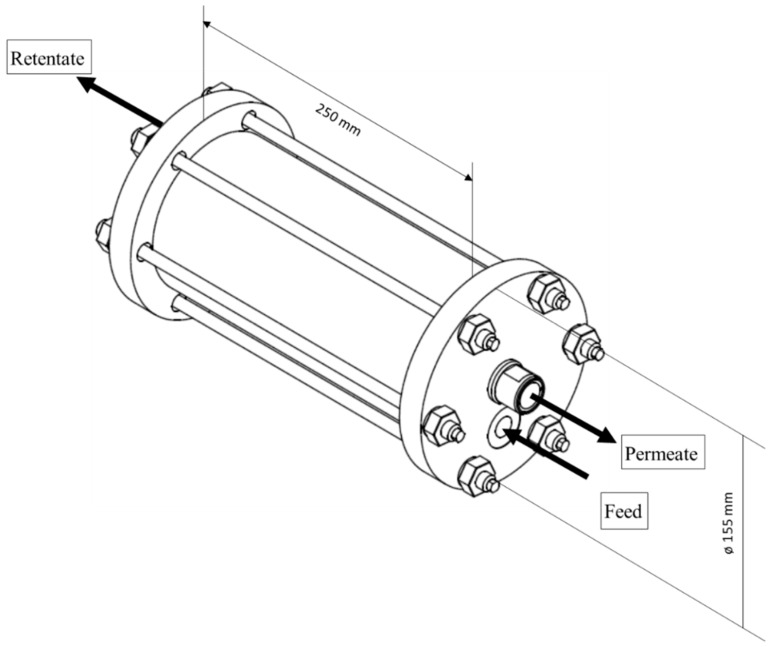
Schematic drawing of K100 envelope-type flat-sheet membrane module.

**Figure 5 membranes-09-00035-f005:**
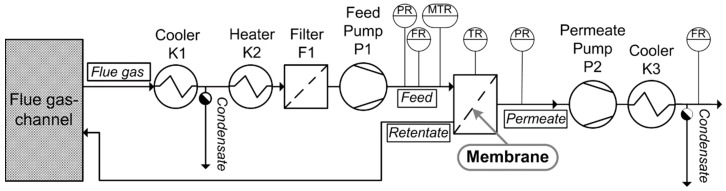
Simplified flow sheet of the single membrane test rig.

**Figure 6 membranes-09-00035-f006:**
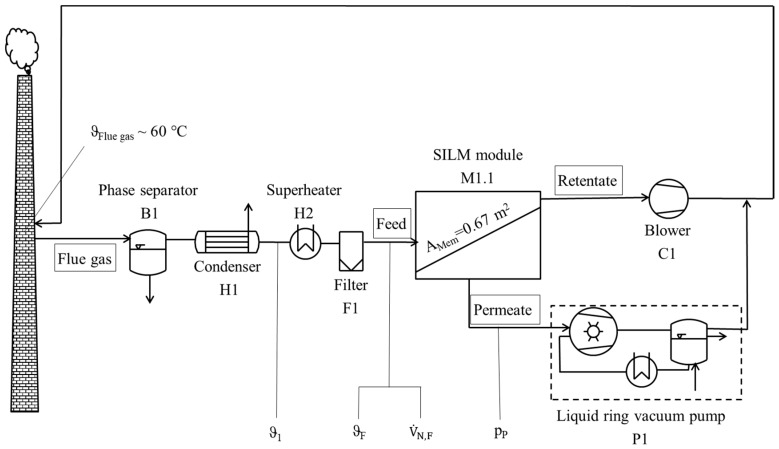
Simplified flow sheet of gas permeation pilot plant.

**Figure 7 membranes-09-00035-f007:**
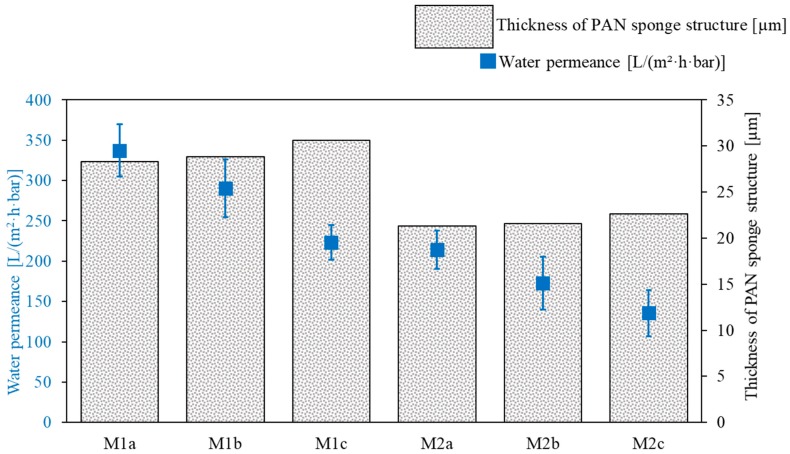
Water permeance and thickness of the PAN membranes M1a–M1c and M2a–M2c used as supports for SILMs.

**Figure 8 membranes-09-00035-f008:**
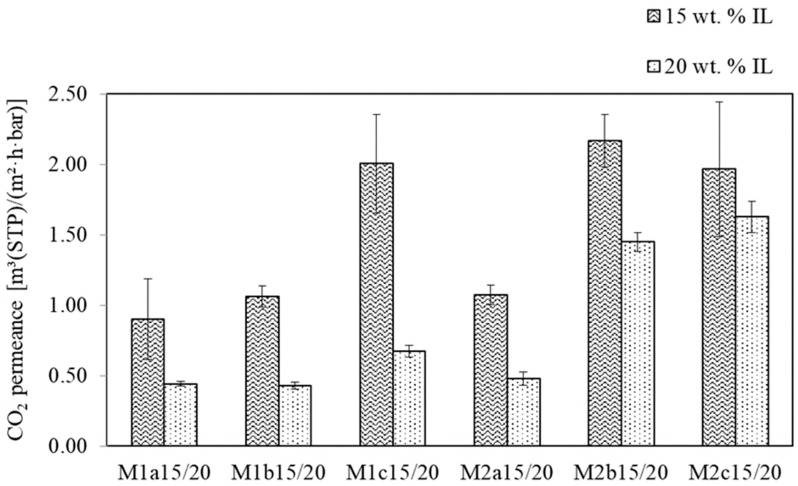
CO_2_ single gas permeances of M1/2a15/20–M1/2c15/20 at ϑ = 28 °C and ∆p = 2 bar.

**Figure 9 membranes-09-00035-f009:**
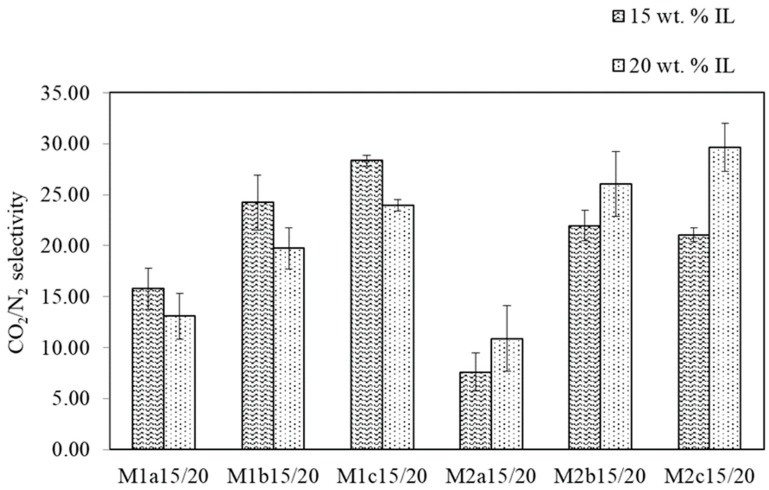
CO_2_ single gas selectivities of M1/2a15/20–M1/2c15/20 (ϑ = 28 °C, ∆p = 2 bar).

**Figure 10 membranes-09-00035-f010:**
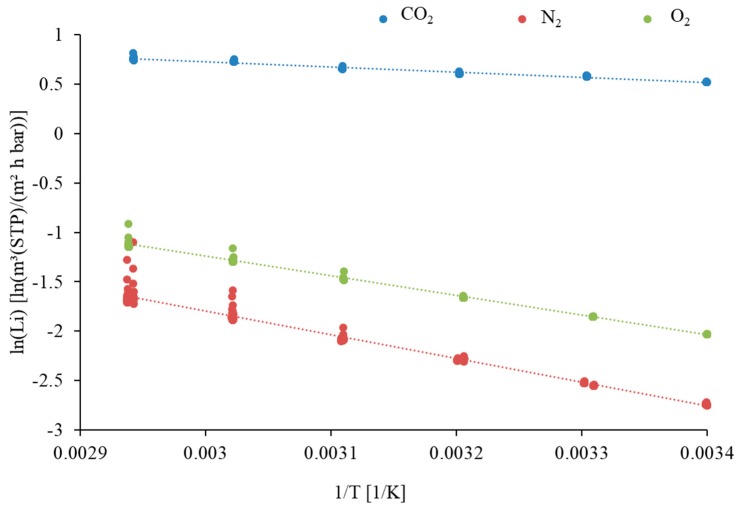
Arrhenius plot of the temperature depending single gas permeances of CO_2_, N_2_ and O_2_ of SILM M1c15 (average pressure ~250 mbar).

**Figure 11 membranes-09-00035-f011:**
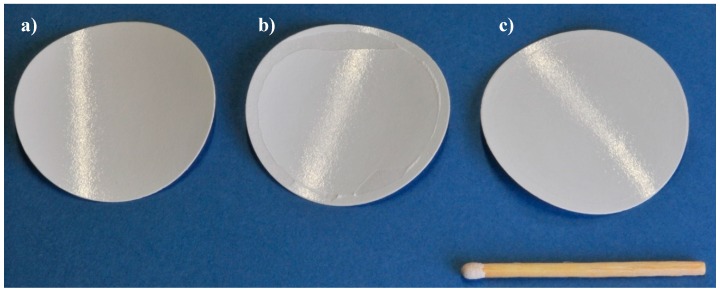
Samples used for the two experiments; (**a**) new sample; (**b**) sample 1 after exposure for 113 h; (**c**) sample 2 after exposure for 335 h.

**Figure 12 membranes-09-00035-f012:**
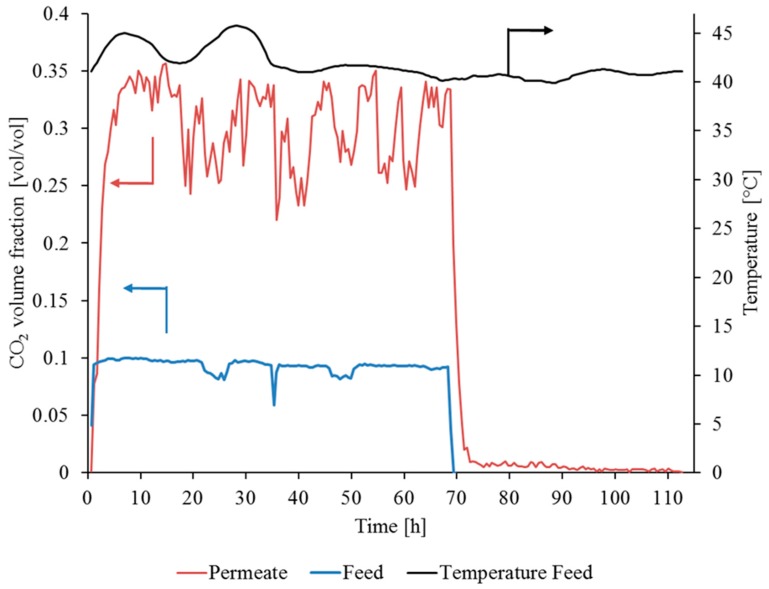
Operation of SILM sample 1 in flue gas: $y_{F,{\mathrm{CO}}_{2}}$, ϑ_F_, ${\dot{V}}_{N,F}$ ~0.12 m^3^(STP)/h, $p_{p}$ ~20 mbar.

**Figure 13 membranes-09-00035-f013:**
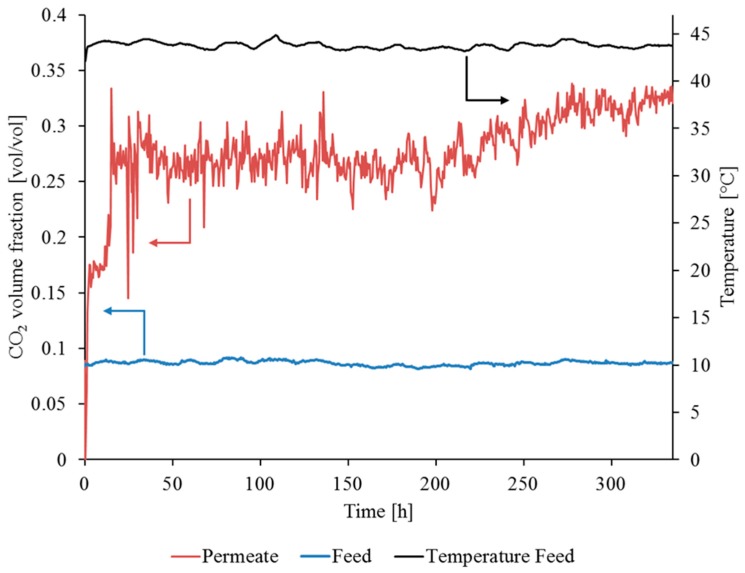
Stable operation of SILM sample 2 in flue gas: $y_{F,{\mathrm{CO}}_{2}}$, $y_{P,{\mathrm{CO}}_{2}}$, ϑ_F_, ${\dot{V}}_{N,F}$ ~0.12 m^3^(STP)/h, $p_{p}$ ~20 mbar.

**Figure 14 membranes-09-00035-f014:**
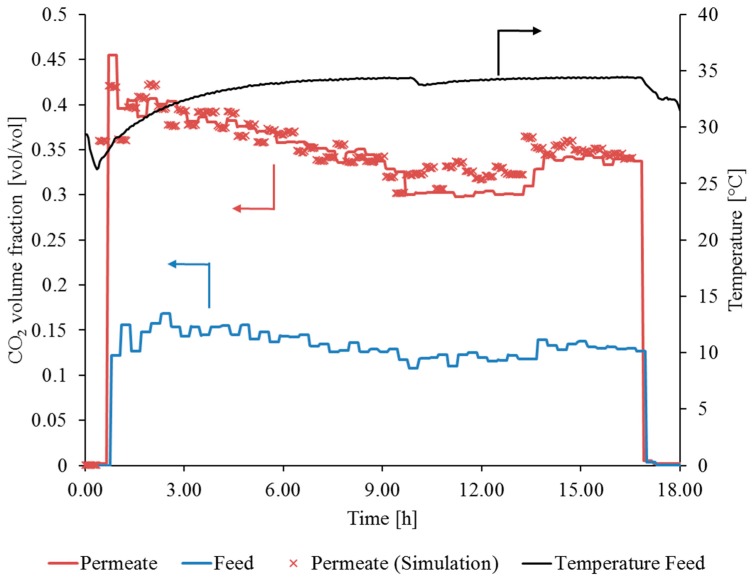
Stable operation of SILM in flue gas: $y_{F,{\mathrm{CO}}_{2}}$, $y_{P,{\mathrm{CO}}_{2}}$, ϑ_F_ during first flue gas operation period, ${\dot{V}}_{N,F}$ ~5.5 m^3^(STP)/h, $p_{p}$ ~125 mbar.

**Figure 15 membranes-09-00035-f015:**
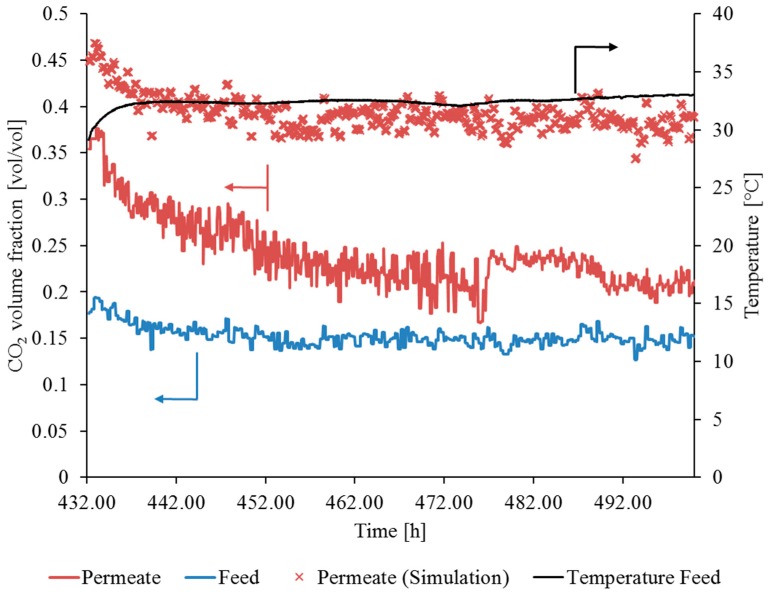
SILM failure during flue gas operation: $y_{F,{\mathrm{CO}}_{2}}$, $y_{P,{\mathrm{CO}}_{2}}$, ϑ_F_ during second flue gas operation period, ${\dot{V}}_{N,F}$ ~5.5 m^3^(STP)/h, $p_{p}$ ~125 mbar.

**Figure 16 membranes-09-00035-f016:**
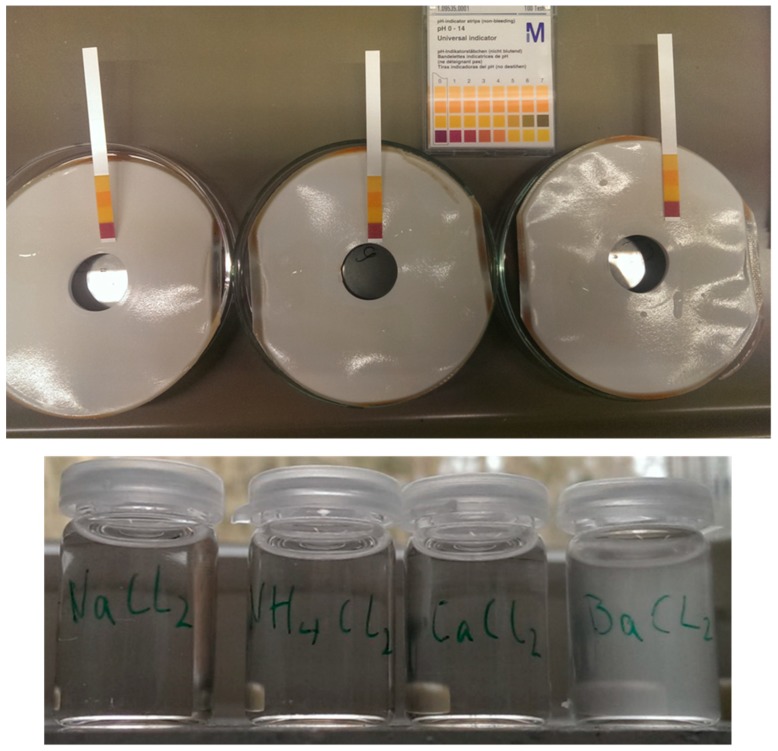
Acidic liquid on SILM envelopes (**top**) proven to contain SO_4_^2−^ (white turbidity (BaSO_4_) in BaCl_2_) (**bottom**).

**Figure 17 membranes-09-00035-f017:**
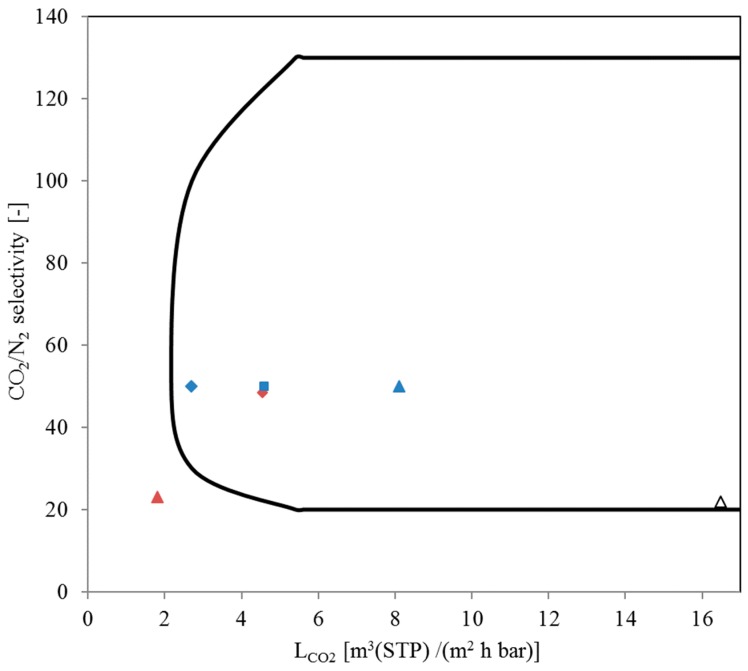
CO_2_/N_2_ selectivity vs. CO_2_ permeance trade-off plot [44]/Comparison of IL membranes with polymeric membranes: ♦ Polaris™ Gen-1 (commercial scale) [11]; ■ Polaris™ Gen-2 (pilot scale) [11]; ▲ Polaris™ advanced (lab scale) [11]; ♦ PolyActive™ 1500 [9]; ▲ SILM with EMIM Tf2N ∆ TFCM with EMIM Tf2N [21].

**Table 1 membranes-09-00035-t001:** Ionic liquid used in this study and its physical properties.

Abbreviation	Structure	Molecular Mass [g/mol]	Viscosity at 25 °C [mPa·s]	Density at 25 °C [kg/m^3^]
EMIM Tf2N	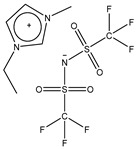	391.3	32 [24]	1518 [25]

**Table 2 membranes-09-00035-t002:** Polyacrylonitrile (PAN) membranes used as support for supported ionic liquid membranes (SILMs): differences in preparation conditions and resulting porous layer thickness.

PAN Membrane	Solvent Variant	Doctor Blade Gap Width [µm]	PAN Thickness [µm]
M1a	1	150	28.3
M1b	1	170	28.8
M1c	1	200	30.6
M2a	2	150	21.3
M2b	2	170	21.6
M2c	2	200	22.6

**Table 3 membranes-09-00035-t003:** Arrhenius model parameters of CO_2_, O_2_ and N_2_ (temperature range of 21–67 °C).

Component	L_i_ at 33 °C [m^3^(STP)/(m^2^·h·bar)]	$L_{\infty ,i}^{0}$ [m^3^(STP)/(m^2^·h·bar)]	E_i_ [kJ/mol]
CO_2_	1.82	6.21	3.122
O_2_	0.17	74.00	15.439
N_2_	0.09	206.00	19.766

**Table 4 membranes-09-00035-t004:** Average CO_2_ permeance, CO_2_/N_2_ selectivity and O_2_/N_2_ selectivity of SILM envelopes (ϑ = 20 °C, ∆p = 2 bar).

Total Membrane Area [m^2^]	L_CO2_ [m^3^(STP)/(m^2^·h·bar)]	α_CO2/N2_ [-]	α_O2/N2_ [-]
0.67	1.53 ± 0.35	22.97 ± 1.54	1.79 ± 0.34

**Table 5 membranes-09-00035-t005:** Results of SILM module quality control with compressed air and single gases (ϑ = 25 °C, ∆p = 2 bar).

Feed Gas for Quality Control	L_O2_ [m^3^(STP)/(m^2^·h·bar)]	L_N2_ [m^3^(STP)/(m^2^·h·bar)]	α_O2/N2_ [-]
Compressed air	0.12	0.09	1.33
Single gases	0.13	0.08	1.63

**Table 6 membranes-09-00035-t006:** Single gas measurements of M1c15 sample 1 and sample 2 performed before and after the experiment in the flue gas (ϑ = 35 °C, ∆p = 3 bar).

Parameter	Sample 1		Sample 2	
	Before Experiment	After Experiment	Before Experiment	After Experiment
CO_2_ permeance [m^3^(STP)/(m^2^·h·bar)]	2.43	2.68	2.46	2.90
N_2_ Permeance [m^3^(STP)/(m^2^·h·bar)]	0.09	0.16	0.11	0.45
CO_2_/N_2_ selectivity [-]	27.00	17.11	22.36	6.44

**Table 7 membranes-09-00035-t007:** Results of SILM envelope test before and after flue gas operation (ϑ = 20 °C, ∆p = 2 bar).

Component	CO_2_	O_2_	N_2_
Average permeance before flue gas operation [m^3^(STP)/(m^2^·h·bar)]	1.53 ± 0.35	0.12 ± 0.03	0.07 ± 0.02
Average relative change of permeance after flue gas operation [%]	−41 ± 8	41 ± 40	131 ± 84

**Table 8 membranes-09-00035-t008:** Comparison of coal type and flue gas cleaning steps of both power plants and the influence on formation of SO_3_ aerosols.

Power Station	Coal Type	SCR	ESP	FGD
RWE Niederaussem (block K)	Rhenish low-sulphur lignite	NO (reduction of NO_x_ by combustion measures)	YES	WFGD and FGDplus pilot plant (∆T = 105–67 °C [38])
EnBW Karlsruhe (block 8)	Different hard coals	YES	YES	WFGD (∆T = 140–65 °C)
Influence on formation of SO_3_ aerosols	- the higher the content of sulphur, the higher the percentage of SO_2_ and thus SO_3_ in the flue gas	- catalytic enhancement of SO_2_ oxidation to sulphur trioxide (SO_3_) [42] - due to the reducing agent, ammonia (NH_3_), ammonium sulphate and/or bisulphate may be formed [42]	- the oxidation of SO_2_ to SO_3_ could occur as a result of ozone formation [38]	- sharp cooling of the flue gas in the FGD causes aerosol formation by heterogeneous or homogeneous nucleation due to supersaturation [43]

**Table 9 membranes-09-00035-t009:** Concentration of NH_4_^+^ and SO_4_^2−^ in the membrane washing solutions.

SILM	c(SO_4_^2−^) [mg/L]	c(NH_4_^+^) [mg/L]
Non-flue-gas-exposed	<0.1	0.54
Flue-gas-exposed	<0.1	10.4

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
