# Peer review of "Separation of Carbon Dioxide from Real Power Plant Flue Gases by Gas Permeation Using a Supported Ionic Liquid Membrane: An Investigation of Membrane Stability"

_membranes, 2019, doi:10.3390/membranes9030035_

Round 1
Reviewer 1 Report
General comments
The aim of the paper is to study the feasibility of separating CO2 from post-combustion gases at pilot plant scale using supported ionic liquid membranes. To the best of my knowledge, the work is original and appropriate for the journal scope and readers. The authors report on the preparation of large scale PAN porous supports, immobilization of ILs by dip coating, membrane envelop preparation and eventually, building of a flat sheet SILM module for pilot plant testing. The authors study the influence of a large number of variables on the stability and performance of the resulting SILMs. Unfortunately, this lead to large methods and results sections that are difficult to follow at some points. In addition, the work requires extensive language editing. Therefore, I believe it is important that the paper is improved with the following comments:
1- Extensive language editing is required to correct grammatical errors, bad use of comas, large sentences, etc. Recurrent use of emphatic present and past tenses should be avoided (do show, did increase, did occur, did investigate, did result…).
2- It would be more convenient to present and discuss fig 1 in section 3.1.1.
3- Please clarify if the constant volume / variable pressure method used for permeability measurements is the time-lag method.
4- The authors should indicate that alpha in Eq. (3) represents an ideal gas selectivity calculated from pure gas permeances.
5- In fig. 6, I think the membrane area should be 0.67 m2 instead of 0.64 m2. Also, this work only employes one membrane module, thus including the second parallel module (M1.2) in the figure and text is unnecessary and leads to confusion.
6- In section 3.1.1., please clarify the technique employed to prepare the PAN porous support (phase inversion?) and if possible provide some general information about the solvent phase. Table 2 is not required as the membrane thickness is already reported in figure 7 and the experimental conditions can be explained in the text.
7- Regarding the temperature influence on gas permeability (Fig. 10), it is usually convenient to show this effect in the form of an Arrhenius plot (log P vs 1/T) to see if data fit to a straight line whose slope is the activation energy. Also, the activation energy is usually expressed in kJ/mol. A comparison of your results with published data would be interesting.
8- In Table 5, how do the authors calculate the permeance of O2 and N2 from mixed gas feed (compressed air) using a permeate flow meter?
9- In section 3.3.1, the authors state that the performances of samples 1 and 2 changed after the experiment due to loss of IL from the porous support (line 491), however a loss of IL would have caused a drastic decrease of the selectivity and much higher permeances. It would be interesting to know the water content of the flue gas after the pretreatment steps, probably there is still some moisture. Although the IL employed is rather hydrophobic, it still captures water to some extent and that may lead to the observed gas permeance increase and selectivity decrease.
10- In fig 14 and 15, simulated results are usually presented using solid lines rather than symbols.
11- Fig 17 should be presented and further discussed out of the conclusions section. What does the black solid line represent? The authors should also indicate the experimental conditions at which those results were obtained.
Minor comments
1- The authors may consider writing the ionic liquid abbreviation (EMIM Tf2N) as [C2mim][Tf2N]. Also, the anion name should read bis(trifluoromethylsulfonyl)imide.
2- The authors should carefully review the paper to correct the style of formulas and units: many chemical compounds are writing without subscripts (e.g., SO3, CO2, H2SO4 and many others), also “ºC” should be separated from the numerical value (Table 1 and other).
Author Response
Dear reviewer,
i uploaded a word document with our answers to your comments.
Kind regards,
Patrik Klingberg

Reviewer 2 Report
This paper presented a method of successfully transferring the lab-scale SILM to square-meter-surface-area pilot scale membranes. The membranes were subsequently tested in real life power plants and were found to be stable although the presence of SO3 aerosols may impact the performance. The paper documented lots of data. It is of great importance for practical applications of SILM and the future development of SLIM.
minor edition may help perfectionating this paper. For example, there are some typo's in the manuscript, e.g. in line 288, "mechanical stable" probably should be "mechanically stable". For Fig. 12-15, it would be great if the authors could convert the data to permeance/selectivity and provide a plot for membrane's selectivity and CO2 permeance vs the hours tested (this is optional but the authors don't have to) . I recommend the paper to be accepted after minor edition.
Author Response
Dear reviewer,
we uploaded our answers to your comments.
Kind regards,
Patrik Klingberg

Round 2
Reviewer 1 Report
I think that the manuscript has improved after the review process and it is suitable for publication after correcting the following mistakes:
1) In table 3, the units of Ei are now kJ/mol instead of kJ/kmol
2) the chemical formula of ammonium sulfate, (NH4)2SO4, is not written correctly thoughout the manuscript
3) I still do not understand the meaning of the black solid line in fig. 17, it might be a processing error of the figure when creating the PDF.
Author Response
...
